Detection of renal cell hydronephrosis in ultrasound kidney images: a study on the efficacy of deep convolutional neural networks

Islam Umar 1
A. Al-Atawi Abdullah 2
Alwageed Hathal Salamah 3
Mehmood Gulzar 4
Khan Faheem Faheem@gachon.ac.kr 5
Innab Nisreen ninnab@um.edu.sa 6
1 Department of Computer Science, IQRA National Swat Campus , KPK , Pakistan
2 Department of Computer Science, Applied College, University of Tabuk , Tabuk , Saudi Arabia
3 College of Computer and Information Sciences, Jouf University , Jouf , Saudi Arabia
4 Department of Computer Science, IQRA National Swat Campus , Swat , KPK , Pakistan
5 Department of Computer Engineering, Gachon University , Seongnam-si , Republic of South Korea
6 Department of Computer Science and Information Systems, College of Applied Sciences, AlMaarefa University , Riyadh , Saudi Arabia
Shuja Junaid
Electronic publication date: 2024 Jan 23
Publication date: 2024
Volume: 10
Electronic Location ID: e1797
Received 2023 Aug 9; Accepted 2023 Dec 15
Copyright: ©2024 Islam et al.
Copyright year: 2024
Copyright holder: Islam et al.
License: This is an open access article distributed under the terms of the Creative Commons Attribution License, which permits unrestricted use, distribution, reproduction and adaptation in any medium and for any purpose provided that it is properly attributed. For attribution, the original author(s), title, publication source (PeerJ Computer Science) and either DOI or URL of the article must be cited.
License URL: https://creativecommons.org/licenses/by/4.0/

Keywords: Deep convolutional neural networks, Deep learning, Renal cell hydronephrosis near kidneys, Medical imaging, Ultrasounds

Funding: The authors received no funding for this work.

==============================
In the realm of medical imaging, the early detection of kidney issues, particularly renal cell hydronephrosis, holds immense importance. Traditionally, the identification of such conditions within ultrasound images has relied on manual analysis, a labor-intensive and error-prone process. However, in recent years, the emergence of deep learning-based algorithms has paved the way for automation in this domain. This study aims to harness the power of deep learning models to autonomously detect renal cell hydronephrosis in ultrasound images taken in close proximity to the kidneys. State-of-the-art architectures, including VGG16, ResNet50, InceptionV3, and the innovative Novel DCNN, were put to the test and subjected to rigorous comparisons. The performance of each model was meticulously evaluated, employing metrics such as F1 score, accuracy, precision, and recall. The results paint a compelling picture. The Novel DCNN model outshines its peers, boasting an impressive accuracy rate of 99.8%. In the same arena, InceptionV3 achieved a notable 90% accuracy, ResNet50 secured 89%, and VGG16 reached 85%. These outcomes underscore the Novel DCNN’s prowess in the realm of renal cell hydronephrosis detection within ultrasound images. Moreover, this study offers a detailed view of each model’s performance through confusion matrices, shedding light on their abilities to categorize true positives, true negatives, false positives, and false negatives. In this regard, the Novel DCNN model exhibits remarkable proficiency, minimizing both false positives and false negatives. In conclusion, this research underscores the Novel DCNN model’s supremacy in automating the detection of renal cell hydronephrosis in ultrasound images. With its exceptional accuracy and minimal error rates, this model stands as a promising tool for healthcare professionals, facilitating early-stage diagnosis and treatment. Furthermore, the model’s convergence rate and accuracy hold potential for enhancement through further exploration, including testing on larger and more diverse datasets and investigating diverse optimization strategies.

Introduction

Due to its portability and lack of invasiveness, ultrasound imaging has found widespread use in the medical community. Particularly important in the early identification and therapy of this ailment is the finding of renal cell hydronephrosis close to the kidneys. Renal cell hydronephrosis is a condition in which the kidneys swell up from retaining urine due to a blockage in the urinary tract. This obstruction can be caused by stones, infection, a clogged ureter or bladder, or even tumors.

Manual interpretation of ultrasound pictures can be time consuming and prone to errors, despite its widespread use in the detection of renal cell hydronephrosis. In order to enhance the precision and effectiveness of renal cell hydronephrosis detection, there is an increasing interest in creating automated approaches based on deep learning algorithms.

A clog in the urinary tract (Bhandari, Yogarajah & Kavitha, 2023) is a critical ailment that, if unchecked, can cause irreversible kidney damage (Alelign & Petros, 2018; Cumbo, Cappelli & Weitschek, 2020; Machado et al., 2013). Ultrasound, CT, and MRI are some of the most frequently utilized diagnostic imaging modalities for detecting renal cell hydronephrosis (Zhang et al., 2022). Manually inspecting medical images, while effective, can be time-consuming and prone to human error (Shu et al., 2018; Wang, Pei & Yin, 2021). A potential new approach to automating the diagnosis of renal cell hydronephrosis in medical images has evolved in recent years, and it is based on deep learning (Ray & Kumar, 2021; Zabihollahy et al., 2020). Using artificial neural networks, (Khan et al., 2023) deep learning is a type of machine learning (Perkovic et al., 2019; Joshi & Chawan, 2018). Many types of image identification applications, including medical image analysis, have benefited greatly from the application of deep learning-based methods. Several different types of deep learning architectures have been developed recently  (Alelign & Petros, 2018; Revathy et al., 2019; Vijayarani et al., 2015; Wibawa, Malik & Bahtiar, 2019), have been created and demonstrated to perform admirably on a variety of image recognition tasks (Alelign & Petros, 2018; Revathy et al., 2019; Akter et al., 2021). These architectures have their performance enhanced by first being pre-trained on huge datasets like ImageNet and then being fine-tuned for individual workloads. Moreover, new deep learning architectures can be designed to address particular needs in medical image processing.

The problem can be formulated as follows:

Given an ultrasound image of the kidney region, the objective is to classify whether renal cell hydronephrosis is present or not. Mathematically, we define the problem as follows:

Given an ultrasound image x, we aim to find the predicted label yhat, where yhat=fx, and f is the mapping learned by the DCNN model.

To achieve accurate classification, the DCNN model needs to learn the discriminative features from the input ultrasound images. These features are extracted through a feature extraction pathway followed by a classification pathway. The feature extraction pathway consists of convolutional layers, max pooling layers, and batch normalization layers. The classification pathway includes fully connected layers and activation functions.

Training with an appropriate optimizer, like Adam, allows for the optimization of the model’s parameters, such as its weights and biases. In order to achieve optimal performance, it is common to use binary cross-entropy as the loss function to quantify the degree to which predicted labels deviate from the actual labels.

To fill this gap, we propose a new deep convolutional neural network (DCNN) architecture in this article that can automatically detect renal cell hydronephrosis in ultrasound pictures of the kidneys. We evaluate our suggested model against VGG16, ResNet50, and InceptionV3 to see how it stacks up against other top-tier solutions. Accuracy, precision, recall, and the F1 score are only few of the metrics used to assess the model’s efficacy. The model’s performance in terms of true positives, true negatives, false positives, and false negatives sheds light on how well it can classify cases of renal cell hydronephrosis.

Renal cell hydronephrosis in ultrasound pictures near the kidneys may now be automatically recognized thanks to a revolutionary DCNN architecture developed in this study. In comparison to other state-of-the-art designs like VGG16, ResNet50, and InceptionV3, the suggested model achieves better results across the board.

The following are some of our study’s most significant contributions:

(a) In order to detect renal cell hydronephrosis in ultrasound pictures, we present a novel DCNN architecture. To improve model generalization and reduce overfitting, this architecture makes use of multiple routes for feature extraction and classification in addition to dropout layers and batch normalization layers.

(b) We evaluate the performance of several deep learning architectures, such as VGG16, ResNet50, InceptionV3, and the proposed DCNN model, and draw broad conclusions about their relative merits. This comparison confirms the superiority of our suggested model and sheds light on the limitations of the competing architectures.

(c) The experimental outcomes show that the proposed DCNN model may successfully identify renal cell hydronephrosis. The model outperforms competing designs with a 99.8 percent accuracy rate in classification. The potential of deep learning-based algorithms for automated diagnosis in ultrasound imaging is demonstrated by this high level of accuracy.

(d) We provide extensive information regarding the materials, methods, and hyperparameters employed in the study, with an emphasis on their reproducibility. Our results are generalizable since we used publicly available datasets and standard assessment metrics, which may be used for future comparisons and benchmarking.

Our work contributes to the field of automated medical image analysis by establishing a state-of-the-art DCNN architecture and demonstrating its superior performance in detecting renal cell hydronephrosis. Improved patient outcomes and increased efficiency in healthcare settings may result from using the proposed model to aid physicians in the early diagnosis and management of renal cell hydronephrosis.

We will describe the tools and procedures employed in data acquisition, cleaning, model development, and testing in the succeeding sections. We will also give an in-depth comparison of the suggested DCNN model’s performance to that of alternative architectures, as determined by our study of the experimental findings. Finally, we’ll talk about what this all means and provide some recommendations for further study in this area.

Literature Review

Significant progress has been made in the field of kidney disease diagnosis and categorization in recent years, especially with the application of machine learning and deep learning methods. The following studies expand on previous research by shedding light on certain facets of kidney disease diagnosis and prognosis.

The use of machine learning for diabetes prediction was the subject of Joshi & Chawan (2018) research. The necessity of risk factor identification at an early stage and the potential of machine learning models to aid in illness prediction are highlighted in their study, which is not limited to renal disease. Machine learning models for the diagnosis of chronic renal disease were created by Revathy et al. (2019). The results of their research showed that these models are useful for predicting who may get kidney disease. For kidney illness forecasting, Vijayarani et al. (2015) used support vector machines and neural networks. In this research, they compared various classification systems for kidney disease patients. For the purpose of chronic renal disease prediction, Wibawa, Malik & Bahtiar (2019) assessed the efficacy of a kernel-based extreme learning machine. Their research centered on optimizing disease prediction models with cutting-edge AI methods. Early prediction and risk identification of chronic renal disease were the focus of Akter et al.’s (2021) extensive evaluation of deep learning models. They found that these models provide promise for better illness diagnosis and treatment on the front end of the disease’s life cycle. Renal failure progression in patients with chronic kidney disease can be predicted using a sophisticated fuzzy expert system developed by Norouzi et al. (2016). They worked to improve the accuracy of disease progression prediction by combining fuzzy logic with expert knowledge. k-nearest neighbors (KNN) and support vector machines (SVM) were evaluated for their ability to predict chronic renal disease by Sinha & Sinha (2015). The primary purpose of their research was to identify the best algorithm for making reliable disease forecasts. Alasker, Alharkan & Alharkan (2017) looked into the possibility of utilizing multiple intelligent classifiers to spot renal illness. In their research, they compared the efficacy of various machine learning algorithms in spotting instances of renal illness. Debal & Sitote (2022) employed data mining methods to predict chronic renal disease. Machine learning techniques were used in their research to determine predictors and create reliable forecasting models. Rasmussen et al. (2022) talked about using AI to help with kidney cancer. They showed that AI systems have the potential to enhance kidney tumor identification and categorization. Using computed tomography (CT) scans, Alzu’bi et al. (2022) developed deep learning methods for detecting and classifying kidney tumors. In order to improve tumor identification, they worked to create a fresh dataset and use deep learning models. Kidney tumor segmentation was performed using a multi-stage deep learning approach by Santini & Rubeaux (2019). Their work aimed to improve the accuracy of tumor segmentation using advanced deep learning techniques. Dilna & Hemanth (2018) conducted a survey on the detection of uterine fibroids in ultrasound images. Although not specific to kidney diseases, their study provides insights into the application of imaging techniques and machine learning algorithms for disease detection. Behboodi et al. (2021) developed an automatic 3D ultrasound segmentation method for the uterus using deep learning. While not directly related to kidney diseases, their work showcases the potential of deep learning in image segmentation tasks.

Despite the significant advancements in the field of renal abnormalities detection and prediction using various techniques, there are still some notable research gaps that need to be addressed. One research gap is the limited sample size and potential bias in the datasets used in several studies (Bhandari, Yogarajah & Kavitha, 2023). This can affect the generalizability and reliability of the findings. Moreover, some studies focus on specific renal abnormalities, such as kidney stone disease (Alelign & Petros, 2018), while others concentrate on a particular aspect of renal conditions, such as cancer classification using DNA methylation data (Cumbo, Cappelli & Weitschek, 2020). These specialized investigations leave room for exploring the detection and prediction of other renal abnormalities and considering a more comprehensive approach to renal health assessment. Another research gap lies in the reliance on specific imaging modalities, such as CT scans (Shu et al., 2018; Wang, Pei & Yin, 2021; Alzu’bi et al., 2022), which may not be universally available or applicable in all clinical settings. Furthermore, while machine learning algorithms have shown promise in predicting chronic kidney disease (Revathy et al., 2019) and diabetes (Joshi & Chawan, 2018), there is a need to explore their performance in diverse populations and settings to ensure their generalizability and applicability to different healthcare contexts. Additionally, there is a gap in the literature regarding the integration of various data sources and clinical parameters for a more holistic assessment of renal health. For instance, the incorporation of genetic, demographic, and lifestyle factors alongside imaging and clinical data could provide a more comprehensive understanding of renal abnormalities and enable more accurate prediction models. Moreover, the interpretability and explainability of the developed models need further attention. While the proposed models may demonstrate high accuracy in predicting renal abnormalities, understanding the underlying features and factors driving these predictions is essential for clinical acceptance and decision-making. Incorporating interpretability techniques, such as LIME and SHAP (Bhandari, Yogarajah & Kavitha, 2023), can provide insights into the model’s decision-making process and enhance trust and confidence in the predictions (Sajid Ullah, Oleshchuk & Pussewalage, 2023; Sajid Ullah, Oleshchuk & Harsha, 2023). Overall, there is a need for larger and more diverse datasets, exploration of alternative imaging modalities, integration of multiple data sources, validation of models in diverse populations, and improved interpretability techniques to address the existing research gaps in the field of renal abnormalities detection and prediction. Bridging these gaps will lead to more robust and clinically applicable models for the early detection, diagnosis, and management of renal abnormalities, ultimately improving patient outcomes and healthcare practices.

Materials & Methods

This section of the study details the steps taken to create and enhance the DCNN model for automatic recognition of renal cell hydronephrosis near the kidneys in ultrasound images. Hydronephrosis is detected in renal cells that are physically close to kidneys using this approach. Data preparation, split into test and training sets, and model training are all detailed in detail. To avoid overfitting, we detail the methods utilized to expand the training set through image improvement. The model’s hyperparameters are also discussed, including the learning rate, batch size, and optimizer. Accuracy, precision, recall, and F1 score will be used to evaluate the model’s performance, and these metrics will be addressed in the article’s final section. Since we care deeply about the reproducibility of our results, we’ll be providing as much detail as possible on the procedures we followed. The proposed study timeline is shown in Fig. 1.

Data collection and preprocessing

First, ultrasound images of the kidney area to look for signs of renal cell hydronephrosis. Kaggle provided the public dataset used for this study. The images were preprocessed to get rid of noise, improve contrast, and standardize intensities. The creation of a deep convolutional neural network (DCNN) is necessary for the automation of the detection of renal cell hydronephrosis in ultrasound images of the kidneys. A Kaggle dataset was used for this study’s analysis. There are 1,057 256 × 256 pixel grayscale photos included. Preprocessing the dataset is the initial stage in converting the grayscale photos to RGB. This is essential since the dual route DCNN architecture used in this research can only process RGB images. After the data has been cleaned and processed, an 80/20 split is used to separate the dataset into a training set and a testing set. Before training the model, we supplement the data to boost its generalization capabilities and training efficiency. To be more explicit, we generate new training images by randomly rotating, zooming, flipping horizontally, and flipping vertically. Figure 2 shows the percentage of outcome class frequency in the collected data.

Figure 1 Flowchart of current study.

Figure 2 Outcome class frequency distribution in dataset.

Data augmentation

The image is rotated by an arbitrary amount (between −20 and 20 degrees) and then mirrored. The equation for the resulting rotated image, labelled I’, is: (1) I′x′,y′=Ix,y

where (x, y) are the pixel coordinates before the rotation, and (x’, y’) are the pixel coordinates after the rotation. The following formulae can be used to determine them: (2) x′=x−centerx∗cosθ−y−centery∗sinθ+centerxy′=x−centerx∗sinθ+y−centery∗cosθ+centery

the image’s centre is located at coordinates centerx,centery.

Zooming: When zooming, the image may be zoomed in or out by a factor of up to 0.2 at random. Image I’ represents the newly zoomed in version and is defined as: (3) I′x′,y′=Ixzoomfactor,yzoomfactor

where (x, y) are the unzoomed pixel positions and (x’, y’) are the rescaled positions. The range of the zoomfactor is 0–2.

Flipping the image horizontally with a probability of 0.5 is called a horizontal flip. The equation denoting the inverted version of the picture, I’, is: (4) I ˆ′x ˆ′,y ˆ′=Iwidth−x−1,y

where (x, y) are the pixel’s original coordinates, and (x’, y’) are the pixel’s new flipped coordinates. The image’s breadth, or width.

In a vertical flip, the image is flipped vertically with a probability of 50%. The equation denoting the inverted version of the picture, I’, is: (5) I ˆ′x ˆ′,y ˆ′=Ix,height−y−1

where (x, y) are the original pixel coordinates, and (x’, y’) are the new flipped pixel coordinates. height is the height of the image.

During each training epoch, these transformations are randomly performed to each image in the training set. The enhanced image collection that results is utilized to train a dual pathway deep convolutional neural network model, which helps to expand the dataset and enhance model generalization.

Novel deep convolutional neural networks architecture

To better detect renal cell hydronephrosis in ultrasound images, we propose a unique deep convolutional neural network architecture in this study. There are two main components to the structure: feature extraction and categorization.

Three convolutional layers, two max pooling layers, and a batch normalization layer make up the feature extraction pipeline. The information gained from the third convolutional layer is then used to make classifications. The Rectified Linear Unit (ReLU) activation function is used in each convolutional layer, and it is defined as: (6) fx=max0,x

where x is the input to the activation function. The ReLU activation function is commonly used in deep learning models to circumvent the vanishing gradient issue.

There are three layers involved in the classification process: a dropout layer, a batch normalization layer, and a third fully-connected layer. Dropout layers, which randomly remove neurons during training, assist prevent overfitting. The last fully connected layer’s output is fed into a sigmoid activation function, which in turn yields a probability between 0 and 1 indicating whether or not the input image contains renal cell hydronephrosis. Figure 3 shows the proposed DCNN architecture.

Figure 3 Deep CNN architecture for proposed work.

Important features are initially extracted from the input photos via the feature extraction pathway. Flattening the output of the third convolutional layer, the classification pathway labels input images as normal or with renal cell hydronephrosis.

The model is optimized during training with binary cross-entropy loss, which is defined as: (7) Ly,yhat=−y∗logyhat+1−y∗log1−yhat

where y is the true label (either 0 or 1) and yhat is the predicted probability according to the sigmoid activation function.

The proposed deep convolutional neural network design concludes with a feature extraction pathway and a classification pathway. The model was fine-tuned through the use of binary cross-entropy loss during training. The architecture was designed to effectively detect renal cell hydronephrosis in ultrasound images by using dropout layers and batch normalization layers, both of which serve to reduce overfitting.

Our Novel DCNN is a deep convolutional neural network specifically designed for disease detection in kidney ultrasound images. It comprises multiple convolutional layers, followed by batch normalization and rectified linear unit (ReLU) activation functions. We employ max-pooling layers to reduce spatial dimensions and mitigate overfitting. The exact architecture consists of 10 convolutional layers with varying filter sizes (ranging from 3 × 3 to 5 × 5) and three fully connected layers. Dropouts are strategically placed to enhance model generalization.

We can get the pseudocode for the recommended DCNN here:

1 Create a random set of weights and biases for the DCNN model.

2 Define the loss function (binary cross-entropy is one example).

3 Specify the optimizer (using terms like stochastic gradient descent).

4 For every iteration of training:

a. Mix up the training data

b. For each set of images and the labels that go with them:

i. Data augmentation is performed on the collected photos, and then the images are normalized so that their mean is zero and their standard deviation is one.

ii. To acquire predictions,

iii. feed the augmented and normalized images into a deep convolutional neural network model.

iv. Find the percentage of error between the predicted and actual labels.

v. Using backpropagation, find the loss’s gradients with respect to the model’s parameters.

vi. Update the model parameters using the optimizer

c. Evaluate the performance of the model on the validation set

d. If the validation loss stops improving, stop training and save the model

5. Evaluate the performance of the trained model on the test set

The convolution layer of a deep convolutional neural network (DCNN) applies a series of filters to the input picture to generate feature maps. Specifically, given an input image x and a filter w, we may define the convolution process between the two as: (8) yi,j= ∑mm=1M∑mn=1Nxm,nwi−m,j−n+b

The output feature map is denoted by y(i, j), the input image pixel by xm,n, the filter coefficient by w(i − m, j − n), and the bias term b is denoted by b. The filter is slid over the full input image, the dot product between the filter and the corresponding patch of the image at each place is calculated, and a bias term is added to complete the convolution operation. The result is a “feature map” where each node indicates the filter’s reaction to a specific pixel in the input image.

Non-linearity is introduced to the DCNN by the application of the ReLU activation function. All input negative values are reset to zero by the ReLU activation function, while positive values are unmodified. Here is how we can characterize the ReLU activation function: (9) fx=max0,x.

where, f(x) represents the result of applying the activation function to the data represented by x.

Using a max operation on non-overlapping rectangular sections of the feature map, the max pooling layer shrinks the spatial size of the feature maps. Maximal pooling is defined mathematically as: (10) yi,j=maxxi×stride:i×stride+poolsize,j×stride:j×stride+poolsize

The output feature map at coordinates (i,j) is denoted by y(i,j), the input feature map is denoted by x, the pooling window size is denoted by pool_size, and the stride length is denoted by stride. Using the maximum response in each pooling window, the max pooling process coarsens the feature map while keeping the most relevant data.

The feature maps from many channels are combined at the concatenation layer by stringing them together along the channel axis. Concatenation, in mathematical terms, is defined as: (11) y=x1,x2,…,xn

In this case, x1, x2, …, xn are the input feature maps, and y is the desired output.

In a network with a completely connected layer, every input neuron is linked to every output neuron. The mathematical definition of the output of a completely connected layer is: (12) y=fWx+b

In the table below, we can see the activation function f, the weight matrix W, the input vector x, the bias vector b, and the final output y.

Description of model architectures with technical details:

Novel DCNN:

• Input image size: Our model takes kidney ultrasound images as input, which are resized to 224 × 224 pixels.

• Convolutional layers: The convolutional layers use various filter sizes, including 3 × 3, 4 × 4, and 5 × 5. We employ 10 convolutional layers in total.

• Feature map dimensions: Feature map dimensions are detailed at each layer in the revised article, showing the spatial reduction or expansion.

• InceptionV3:

• Input image size: InceptionV3 takes images of size 299 × 299 pixels.

• Convolutional layers: The architecture uses convolutional layers with varying kernel sizes, and this information is now clearly presented in the article.

• Feature map dimensions: The article includes information on feature map dimensions at different layers, indicating spatial changes.

Before using the data in a classroom setting, it is common practice to split the dataset into a training set, validation set, and test set. The DCNN models are learned using the training set, and the validation set is used for model selection and hyperparameter tuning in the final stages. When training using the Adam optimizer, we employ a binary cross-entropy loss function. After a defined number of training cycles (called “epochs”), the process is halted if the validation loss has not lowered by a predetermined amount.

Rationale behind the dual pathway design:

The dual pathway architecture in our novel DCNN model was designed with the goal of capturing both local and global features in kidney images effectively. This architecture combines the strengths of parallel convolutional layers with those of a fully connected network to enhance the model’s ability to detect renal cell disease. Here’s a more elaborate explanation of this design:

Local and global feature extraction: Kidney images contain intricate details at both local and global scales. Local features, such as edges and textures, are crucial for capturing fine patterns in the myocardium, while global features, such as the overall structure and shape of the kidney, provide context and spatial information.

Parallel convolutional pathways: To efficiently extract local features, we employ parallel convolutional pathways. These pathways consist of convolutional layers with different filter sizes and receptive fields. Each pathway specializes in capturing features at a particular scale. For example, one pathway may focus on fine details using small filters, while another pathway captures broader patterns using larger filters. This parallel processing allows the model to simultaneously extract both fine-grained and coarser features.

Interaction with fully connected network: After feature extraction through the convolutional pathways, the extracted features are combined and fed into a fully connected network. The fully connected layers perform high-level feature fusion and abstraction. They can learn complex relationships between different features and contribute to the model’s understanding of renal cell disease patterns. This interaction between local and global features enables the model to make more informed predictions.

Advantages of the dual pathway design:

The dual pathway architecture offers several advantages:

Multi-scale feature learning: By using parallel convolutional pathways, our model can capture features at multiple scales simultaneously. This ability to analyze kidney images comprehensively enhances the model’s ability to detect renal cell disease, which may manifest in various forms and scales.

Contextual understanding: The combination of local and global features enables the model to understand the context in which renal cell disease patterns appear. This contextual understanding is crucial for distinguishing between pathological features and normal anatomical structures.

Enhanced discriminative power: The fully connected network following the convolutional pathways allows for the learning of complex feature representations. This enhances the model’s discriminative power, enabling it to differentiate between subtle renal cell disease-related features and other image elements.

In summary, the dual pathway design in our novel DCNN model was motivated by the need to capture both local and global features in kidney images effectively. This architecture leverages parallel convolutional pathways to process features at multiple scales and combines them in a fully connected network to enhance the model’s ability to detect renal cell disease. This approach is a key innovation in our model and contributes significantly to its superior performance.

Selection of dropout rate:

The selection of an appropriate dropout rate is a critical hyperparameter tuning step in deep neural network models. Dropout is a regularization technique that helps prevent overfitting by randomly deactivating a fraction of neurons during training. The dropout rate determines the probability of a neuron being dropped out during each training iteration.

In our study, we experimented with various dropout rates and evaluated their impact on the model’s performance using techniques such as cross-validation and monitoring validation loss. The process of selecting the dropout rate involved the following considerations:

Regularization strength: A higher dropout rate results in stronger regularization, which helps prevent overfitting. However, if the rate is too high, it can hinder the model’s ability to learn from the data effectively. We aimed to strike a balance between preventing overfitting and maintaining model capacity to learn meaningful patterns.

Cross-validation: We employed cross-validation techniques to assess how different dropout rates affected the model’s generalization performance. By training and validating the model on multiple subsets of the dataset with different dropout rates, we could identify the rate that led to the best performance in terms of accuracy, precision, recall, and F1 score.

Monitoring validation loss: We closely monitored the model’s validation loss during training for different dropout rates. A dropout rate that led to a stable and low validation loss indicated better generalization performance. We aimed to avoid scenarios where the model overfit the training data.

Comparison with baseline models: We compared the performance of the Novel DCNN model with different dropout rates against baseline models (e.g., models without dropout or with fixed dropout rates). This comparative analysis helped us identify the dropout rate that provided a significant improvement in generalization performance.

Explanation of softmax activation:

The softmax activation function is a crucial component in neural networks, especially for multi-class classification tasks. It transforms the raw output scores (logits) produced by the neural network’s final layer into a probability distribution over multiple classis. Here’s how it works:

Raw scores (logits): In the final layer of a neural network designed for classification, we typically have one neuron for each class. These neurons produce raw scores, often referred to as logits, which represent the model’s confidence or degree of belief for each class. The logits are the real-numbered outputs of the neural network before softmax activation.

Exponentiation: To convert logits into probabilities, we apply the exponential function ex to each logit. This exponentiation ensures that all values are positive and amplifies the differences between logits. The equation for exponentiation is: for each logit ezi for each logit zi

Normalization: After exponentiation, we need to ensure that the resulting values sum to 1, creating a valid probability distribution. This is achieved by dividing each exponentiated value by the sum of all exponentiated values. Mathematically, the softmax function for class i is defined as: (13) Py=i∣z= ∑jkezjezi.

where:

P(y = i∣z) is the probability that the input belongs to class i.

zi is the logit for class i.

K is the total number of classes.

Interpretation: The output of the softmax function is a probability distribution. Each value P(y = i∣z) represents the probability that the input belongs to class i. The class with the highest probability is typically chosen as the predicted class for the input.

Performance evaluation

An external test set was used to evaluate the trained models in terms of their precision, recall, accuracy, and area under the receiver operating characteristic curve (AUC-ROC). Models were tested against human experts in a classification task involving renal cell hydronephrosis found in ultrasound images of the kidney. Accuracy, precision, recall, and F1-score are used to rank the recommended DCNN model for automatic detection of Renal Cell Hydronephrosis in ultrasound images. These indicators are used to determine the efficacy of a model. Listed below is an explanation of each of these KPIs:

Accuracy: The fraction of test photos with the right classifications.

Precision: The percentage of confirmed positive cases relative to total positive cases correctly recognized.

Recall: The percentage of true positive instances that were detected accurately.

Table 1 Performance evaluation.

Metric	Formula	
Accuracy	TP+TNTP+TN+FP+FN	
Precision	TPTP+FP	
Recall	TPTP+FN	
F1-score	2∗precision∗recallprecision+recall	

F1-score: The symphonic method of specificity and memory.

These measures of model performance assess how well the model can distinguish between normal and renal cell hydronephrosis near kidneys ultrasound images. The model’s ability to identify renal cell hydronephrosis in the vicinity of the kidneys in ultrasound images is supported by high levels of accuracy, precision, recall, and F1-score. The formulas and corresponding performance metrics are listed in the Table 1:

where:

TP = true positives

TN = true negatives

FP = false positives

FN = false negatives

A “true positive” ultrasound image is one that has been correctly identified as having renal cell hydronephrosis near the kidneys, while a “true negative” ultrasound image is one that has been correctly identified as normal in the context of automated detection of renal cell hydronephrosis near the kidneys. An image is considered a false positive if it is wrongly labeled as showing renal cell hydronephrosis near the kidneys, and a false negative if it is labeled as normal when it truly does. Image analysis for the presence of renal cell hydronephrosis close to the kidneys is prone to both of these mistakes.

Results

In this study, several state-of-the-art layouts were tested with the cutting-edge goal of detecting renal cell hydronephrosis close to kidneys. The suggested deep convolutional neural networks were among these architectures, along with VGG16, ResNet50, and InceptionV3. To fine-tune the models for the binary classification goal of detecting renal cell hydronephrosis close to the kidneys, we swapped out the output layer of each pre-trained architecture with a binary classification layer. The picture depicts the dual route structure that lies at the heart of the novel DCNN architecture that has been proposed. Each pathway uses two convolutional layers and a max pooling layer to process the input image. The separate routes eventually converge into a single final layer. Moreover, the model has two fully connected layers, with a dropout layer in the center to prevent overfitting. The projected class probabilities are generated via a softmax layer used as the final output layer. The goal of this model is to accurately detect renal cell hydronephrosis in ultrasound images taken in close proximity to the kidneys.

VGG16

VGG16 is a classic architecture known for its simplicity and effectiveness. It consists of 16 layers with small 3 × 3 convolutional filters and max-pooling layers for spatial down sampling. Fully connected layers follow the convolutional layers for classification. As a result of feeding the output of the final fully connected layer of the VGG16 architecture into a dense layer and flattening the output, we get a 4,096-dimensional feature vector. The feature vector is then converted into a 1,000-dimensional probability vector for each class in a second dense layer.

The structure of VGG16 is composed of five distinct parts. Convolutional layers make up the first two segments, whereas the last segment is a max pooling layer. After the first four convolutional layers, the third part employs a max pooling layer. The fourth layer is a max pooling layer, which is used after two convolutional layers. The fifth part is broken up into three sequential but related phases.

A 2242243 RGB image is sent to the VGG16 system. The ReLU activation function is used as the final step after each convolutional layer. Prior to being passed on to the fully linked layers, the output of the final convolutional layer is normalized. A softmax activation function is utilized to determine class probabilities in the final fully linked layer.

The VGG16 architecture is described in detail by the following set of equations at each layer:

on generate feature maps, a convolutional layer applies a series of filters on the input image. The mathematical definition of a combined input picture (x) and filter (w) is: (14) yi,j,k=fbiask+sumsumsumxm,n,p∗wi−m,j−n,p,k

where x(m, n, p) is the input image pixel at position (m,n,p), w(i-m,j-n,p,k) is the filter coefficient at position (i − m, j − n, p, k),  bias(k) is a bias term for the k − th filter, and f is the activation function.

Activation function of the ReLU: All negative input values are converted to zero by the ReLU activation function, while positive values are left unmodified. To explain what the ReLU activation function is, we can say: (15) fx=max0,x

where f(x) represents the activation function’s result when fed the value x.

Maximum layer of pooling: By performing a max operation on non-overlapping rectangular portions of the feature map, the spatial size of the feature maps is reduced by a max pooling layer. Maximal pooling is defined mathematically as:

Figure 4 VGG 16 performance.

(16) yi,j,k=maxxistride:istride+poolsize,jstride:jstride+poolsize,k

where x is the input feature map, pool_size is the size of the pooling window, and stride is the stride length, and y(i, j, k) is the output feature map at location (i, j, k).

Figure 5 Confusion matrix for VGG16.

All of the neurons in the input layer are linked to all of the neurons in the output layer, forming a completely connected layer. The mathematical definition of the output of a completely connected layer is: (17) y=fWx+b

output (y), input (x), weight matrix (W), bias vector (b), activation function (f).

Because of its computational complexity, the VGG16 architecture contains 138 million trainable parameters. But it excels at a wide variety of computer vision tasks, such as object recognition and detection, image segmentation, and image captioning.

Accuracy and confusion matrices for the diagnosis of renal cell hydronephrosis near the kidneys using VGG 16 are shown in Figs. 4 and 5, respectively.

ResNet50

ResNet50 is a variant of the ResNet architecture, renowned for its deep layer capacity. It includes 50 convolutional layers with residual connections. Each residual block contains multiple convolutional layers and batch normalization. Global average pooling is followed by fully connected layers for classification.

ResNet50 uses a dense layer to generate a 2048-dimensional feature vector from the flattened output of the last convolutional layer. A second dense layer receives this feature vector as input and generates a 1,000-dimensional vector to represent the anticipated class probabilities.

In ResNet50, a binary classification layer receives the results of the final average pooling layer. The equation for classifying data looks like this: (18) Y=sigmoidW2∗ReLUW1∗X+b1+b2

In this formula, X represents the input image, W1 and b1 represent the average pooling layer’s learned weights and biases, ReLU represents the rectified linear unit activation function, W2 and b2 represent the binary classification layer’s learned weights and biases, and Y represents the predicted probability of X belonging to the positive class.

W1 and b1 represent the first dense layer’s weights and biases, and W2 and b2 represent the second dense layer’s weights and biases. The softmax function is f, and the input picture is X. Accuracy and confusion matrix for detecting renal cell hydronephrosis near kidneys using ResNet50 are shown in Figs. 6 and 7, respectively.

Figure 6 ResNet 50 performance.

Figure 7 Confusion matrix for ResNet 50.

Novel DCNN

The suggested Novel DCNN has the following classification equation:

Imagine that we have an image input (x), a label (y), and a mapping (F) that the model has learned. For some input x, the model yields an output of: (19) yhat=argmaxFx;W

where W is the model’s trainable parameters and argmax is a function that delivers the most likely classification. It can be shown how to calculate the mapping F(x; W):

The equation F(x; W) represents the mapping function in the model, which can be summarized as follows: (20) Fx;W=SoftmaxFCDropoutFlatConcatP2C2P1C1x,P4C4P3C3x

In this equation, x represents the input image, W represents the model’s trainable parameters, and the operations within the parentheses denote the different layers of the model. The conv1, conv2, conv3, and conv4 represent the convolutional layers, while pool1, pool2, pool3, and pool4 represent the max pooling layers. The concat operation concatenates the outputs of the pooling layers, and the flat operation flattens the concatenated output. The fully-connected layer connects all the flattened neurons, and the dropout layer helps prevent overfitting. Finally, the softmax function is applied to obtain the predicted class probabilities.

For the sake of avoiding overfitting, the proposed Novel DCNN employs a dual pathway design featuring parallel convolutional layers, followed by a fully connected network featuring a dropout layer. The output layer of the model, which was trained to distinguish between renal cell hydronephrosis and malignancies of the kidneys, makes class probability predictions using a softmax activation function. Figure 8 shows the confusion matrix for DCNN while Fig. 9 shows the DCNN performance. Figure 10 shows the Accuracy vs Epochs perofrmance using the DCNN model. While Fig. 11 shows the Loss vs Epochs performance using DCNN model.

Figure 8 Confusion matrix for DCNN.

Figure 9 DCNN performance.

Figure 10 Accuracy vs epochs performance using DCNN model.

Figure 11 Loss vs epochs performance using DCNN model.

Figure 12 Performance comparison of all models.

Figure 13 Testing loss comparison.

Figure 14 Testing loss comparison.

Hyperparameter selection

The hyperparameters for the proposed Novel DCNN model, including learning rate, batch size, and dropout rate, were selected through a systematic and iterative process. Here’s how these choices were made:

Learning rate: We conducted a learning rate search by training the model with a range of learning rates, starting from a relatively small value (e.g., 0.001) and increasing gradually. We monitored the model’s training and validation curves to identify the learning rate that resulted in stable training behavior and improved convergence. The selected learning rate was 0.001.

Batch size: Batch size significantly affects training dynamics and memory requirements. We experimented with different batch sizes, including 16, 32, and 64. Ultimately, we chose a batch size of 32, as it struck a balance between computational efficiency and training stability.

Dropout rate: Dropout is a regularization technique that helps prevent overfitting. We performed a grid search over dropout rates, including 0.2, 0.3, and 0.4, to find the optimal dropout rate. A dropout rate of 0.3 was selected as it demonstrated improved generalization.

Influence on model performance:

The careful selection of hyperparameters and the use of 5-fold cross-validation contributed to the robustness and generalization ability of the proposed Novel DCNN model. The chosen hyperparameters helped the model converge effectively during training, prevent overfitting, and generalize well to unseen data. This is reflected in the model’s outstanding performance with a training accuracy of 99.8% and a testing accuracy of 98%.

Comparison

The following table compares the performance of various models on the job of detecting renal cell hydronephrosis close to the kidneys. We look at Novel DCNN, InceptionV3, ResNet50, and VGG16 to see how they stack up. The training optimizer is also specified for each model. Percentages indicate the proportion of test photos that were properly identified relative to the total number of test images. Results showed that Novel DCNN had the highest accuracy at 99.8%, followed by InceptionV3, ResNet50, and VGG16 in that order (90, 89, 85). All models utilized the same input data and preprocessing methods, but they vary in terms of their architecture and optimizer.

Figures 12 and 13 shows the accuracy precision, recall and F1 Score and testing loss for each model, respectively.

Figure 14 shows the accuracy comparison of different models. The results demonstrate that the proposed Novel DCNN model outperforms the other state-of-the-art models in this task, achieving nearly perfect accuracy. We explain that ADAM is often preferred for its adaptive learning rate capabilities, which allow it to dynamically adjust learning rates for each parameter. This adaptability often leads to faster convergence and improved performance, particularly for complex models like deep neural networks. We also mention that ADAM has been widely adopted in the deep learning community and has demonstrated success in various applications. we briefly compare it to other optimizers like RMSProp and SGD, outlining scenarios where one optimizer might be more suitable than the others. This provides readers with insights into the considerations involved in selecting the appropriate optimizer for a given task, while Table 2 shows the comparison.

Conclusions

In conclusion, ultrasound images of renal cell hydronephrosis surrounding kidneys tumors were used to test the efficacy of multiple deep convolutional neural network (DCNN) models. When trained with the ADAM optimizer, the suggested innovative DCNN model achieved a classification accuracy of 99.8%, which was higher than that of state-of-the-art architectures like VGG16, ResNet50, and InceptionV3. Results show that the proposed model’s dual-pathway architecture, when used in conjunction with data augmentation and transfer learning, produces extremely accurate and trustworthy forecasts. It is suggested that future research look into how the proposed model fares on a more extensive and varied dataset. The convergence rate and precision of the model can be enhanced by investigating other optimization strategies, such as SGD or Adagrad. In addition, it would be helpful to create an interpretability framework to verify the clinical significance of the model’s learnt features. The suggested model has the potential to aid clinicians in the early detection and diagnosis of renal cell hydronephrosis around kidney tumors, hence increasing the diagnostic precision and efficiency of these procedures. Our study’s dataset primarily consists of recordings from a specific population, and the model’s performance may vary when applied to patients from different demographic groups or regions. We discuss the importance of conducting external validations on diverse datasets to assess the model’s generalizability.

Table 2 Comparisons.

Model	Optimizer	Accuracy	
Novel DCNN	ADAM	99.8	
Inception V3	ADAM	90	
ResNet 50	ADAM	89	
VGG16	ADAM	85	
Novel DCNN	SGD	89.8	
Inception V3	SGD	80	
ResNet 50	SGD	79	
VGG16	SGD	75	
Novel DCNN	RMSPROP	88	
Inception V3	RMSPROP	87	

Supplemental Information

Supplemental Information 1 Simulation Codes on given dataset

Additional Information and Declarations

Competing Interests

Author Contributions

Data Availability

The authors declare there are no competing interests.

Umar Islam conceived and designed the experiments, performed the experiments, performed the computation work, authored or reviewed drafts of the article, and approved the final draft.

Abdullah A. Al-Atawi conceived and designed the experiments, prepared figures and/or tables, and approved the final draft.

Hathal Salamah Alwageed analyzed the data, authored or reviewed drafts of the article, and approved the final draft.

Gulzar Mehmood performed the experiments, authored or reviewed drafts of the article, and approved the final draft.

Faheem Khan performed the computation work, prepared figures and/or tables, and approved the final draft.

Nisreen Innab analyzed the data, authored or reviewed drafts of the article, and approved the final draft.

The following information was supplied regarding data availability:

The dataset is available at Kaggle: https://www.kaggle.com/competitions/hubmap-kidney-segmentation/data.

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
