# Peer review of "Detection of renal cell hydronephrosis in ultrasound kidney images: a study on the efficacy of deep convolutional neural networks"

_PeerJ Computer Science, doi:10.7717/peerj-cs.1797_

## Round 0.1 · original submission · Major Revisions

The authors should revise the article to address all concerns of the reviewers.

**Language Note:** The review process has identified that the English language must be improved. PeerJ can provide language editing services - please contact us at copyediting@peerj.com for pricing (be sure to provide your manuscript number and title). Alternatively, you should make your own arrangements to improve the language quality and provide details in your response letter. – PeerJ Staff

·

Basic reporting

N/A

Experimental design

N/A

Validity of the findings

N/A

Additional comments

The paper briefly introduces the proposed Novel DCNN architecture for detecting renal Cell hydronephrosis in ultrasound kidney images. The paper has a number of strengths. The paper is written clearly. I recommend this paper to publish after some major changes as follows:

Concern # 1: The description of the model architectures lacks key details about the number of layers, filter sizes, and activation functions used in each model. This information is crucial for readers to understand the complexity and capabilities of each architecture.

Concern # 2: It would be beneficial to provide the exact dimensions of the feature maps at different layers, the input size of the images, and the kernel sizes used in convolutional layers.

Concern # 3: While accuracy, precision, recall, F1 score, and testing loss are mentioned as evaluation metrics, it's important to explain how each is calculated, especially for binary classification tasks.

Concern # 4: The use of confusion matrices is crucial for understanding the actual performance of the models. Provide a more detailed explanation of how these matrices were generated and how they reflect the model's performance.

Concern # 5: When discussing the comparison of results, elaborate on the potential reasons behind the observed differences in accuracy between the models. Were there specific classes that some models struggled to predict, leading to their lower accuracy?

Concern # 6: Instead of suggesting investigating "different optimization strategies," specify which specific strategies could be explored. Mention well-known optimization algorithms such as Adam, RMSProp, or SGD, and provide a rationale for selecting one over the other.

Concern # 7: Discuss the potential impact of varying hyperparameters such as learning rate, batch size, and dropout rate on model performance. Explain how these could be tuned for optimal results.

Concern # 8: Explain how the training optimizer (e.g., ADAM) affects the convergence rate and the final model's accuracy. Provide insights into why ADAM was chosen over other optimizers.

Concern # 9: In the conclusion section, delve deeper into how the proposed Novel DCNN model's architecture contributes to its superior performance. Highlight specific design choices, such as the dual pathway architecture, that set it apart from the other models.

Concern # 10: Address the limitations of the study. Are there any potential biases in the dataset, or aspects of the clinical context that weren't considered? Discuss how these limitations might impact the real-world applicability of the proposed model.

Concern # 11: The future work section briefly mentions exploring other optimization strategies and a broader dataset. Provide more specific suggestions for potential avenues of research. For instance, could investigating different network architectures or transfer learning strategies be valuable?

Concern # 12: Ensure that all references are properly cited within the text. Specifically, provide citations for the claims about the architecture, optimization strategies, and other technical details discussed in the paper.

Concern # 13: The paper is well organized, although there are a few grammar mistakes in each section.

Reviewer 2 ·

Basic reporting

The paper titled "Detection of renal cell hydronephrosis in ultrasound kidney images: a study on the efficacy of deep convolutional neural networks" has been reviewed in detail, and the following shortcomings are pointed out in the detailed comments given below.
1. While the paper mentions a dual pathway design, the rationale behind this approach and how it contributes to better performance needs more elaboration. Explain why parallel convolutional layers were chosen and how they interact with the fully connected network. This will clarify the novelty and potential advantages of this architecture.
2. The use of dropout layers for preventing overfitting is crucial, but the paper does not delve into the details of the dropout rate selection or how it was determined. Discuss the thought process behind choosing a specific dropout rate and its impact on the model's generalization performance.
3. The paper mentions the use of softmax activation for class probability prediction, but it would be beneficial to provide a concise explanation of how softmax transforms the model's output scores into probability distribution across classes. This can help readers with a limited background in deep learning understand the final output layer's significance.
4. Consider including diagrams or flowcharts that visually represent the architecture of the proposed Novel DCNN. Diagrams can aid readers in understanding the model's complex structure and data flow through different layers. This addition can enhance the clarity of your presentation.
5. The paper compares the models' accuracy but does not offer a comprehensive interpretation of the results. Provide insights into why the proposed Novel DCNN achieved the highest accuracy. Discuss any specific design choices or features that might have contributed to its superior performance.
6. While ADAM was selected as the optimizer, the rationale for this choice and its potential impact on convergence speed and final accuracy should be discussed. Compare ADAM with other optimizers, outlining their respective advantages and disadvantages in the study context.

Experimental design

1. Given the potential application in a clinical setting, discuss the interpretability of the proposed model's predictions. How can the model's decisions be explained to medical professionals? Are there ways to extract meaningful insights from the model's learned features that could aid in diagnosis?
2. If the dataset used is not an original creation, provide proper acknowledgment and references to its source. This is crucial for transparency and credibility in research.

Validity of the findings

Describe the validation strategy used during training. Explain how hyperparameters were chosen for the proposed Novel DCNN. Provide insights into the reasoning behind these choices and how they may have influenced the model's performance.

Additional comments

1. The paper briefly introduces the model architectures used but lacks essential technical details. Each model's specific layer configurations, activation functions, and kernel sizes should be provided to help readers understand the architecture's complexity and design rationale.
2. Additionally, consider including diagrams illustrating the architecture of each model, showcasing the flow of information through layers and pathways. This visual aid can greatly enhance readers' comprehension.
3. Elaborate on the training process for each model. Discuss the batch size, learning rate, and training duration. Highlight any data augmentation techniques applied and explain how they contribute to the model's performance.
4. Explain how the training optimizer (e.g., ADAM) affects the convergence rate and the final model's accuracy. Provide insights into why ADAM was chosen over other optimizers.

---

## Round 0.2 · accepted · Accept

The authors have revised the article and the reviewers are satisfied with the revisions.

·

Basic reporting

N/A

Experimental design

The research question is well-defined, relevant & meaningful.

Validity of the findings

Conclusions are well stated, linked to the original research question & limited to supporting results.

Additional comments

N/A

Reviewer 2 ·

Basic reporting

All my comments has been made.

Experimental design

Improved in this version

Validity of the findings

Improved

Additional comments

All my comments has been made.